# Association of the rs8720 and rs12587 *KRAS* Gene Variants with Colorectal Cancer in a Mexican Population and Their Analysis In Silico

**DOI:** 10.3390/cells12151941

**Published:** 2023-07-26

**Authors:** Martha Patricia Gallegos-Arreola, Asbiel Felipe Garibaldi-Ríos, José Israel Cruz-Sánchez, Luis Eduardo Figuera, Carlos Alberto Ronquillo-Carreón, Mónica Alejandra Rosales-Reynoso, Belinda Claudia Gómez-Meda, Irving Alejandro Carrillo-Dávila, Ana María Puebla-Pérez, Héctor Montoya-Fuentes, Valeria Peralta-Leal, Guillermo M. Zúñiga-González

**Affiliations:** 1División de Genética, Centro de Investigación Biomédica de Occidente (CIBO), Centro Médico Nacional de Occidente (CMNO), Instituto Mexicano del Seguro Social (IMSS), Guadalajara 44340, Jalisco, Mexico; marthapatriciagallegos08@gmail.com (M.P.G.-A.); asbiel.garibaldi4757@alumnos.udg.mx (A.F.G.-R.); luisfiguera@yahoo.com (L.E.F.); alexcd7.ac@gmail.com (I.A.C.-D.); 2Doctorado en Genética Humana, Centro Universitario de Ciencias de la Salud (CUCS), Universidad de Guadalajara (UdeG), Guadalajara 44340, Jalisco, Mexico; 3Especialidad en Oncología Médica, Universidad de Guadalajara (UdeG)/UMAE Hospital de Especialidades, Centro Médico Nacional de Occidente (CMNO), Instituto Mexicano del Seguro Social (IMSS), Guadalajara 44340, Jalisco, Mexico; israelcsnz@gmail.com (J.I.C.-S.); carlos@ronquillo.mx (C.A.R.-C.); 4Oncología Clínica, UMAE Hospital de Especialidades, Centro Médico Nacional de Occidente (CMNO), Instituto Mexicano del Seguro Social (IMSS), Guadalajara 44329, Jalisco, Mexico; 5División de Medicina Molecular, Centro de Investigación Biomédica de Occidente (CIBO), Centro Médico Nacional de Occidente (CMNO), Instituto Mexicano del Seguro Social (IMSS), Sierra Mojada 800, Col. Independencia, Guadalajara 44340, Jalisco, Mexico; mareynoso@hotmail.com (M.A.R.-R.); schlagzeugger@hotmail.com (H.M.-F.); 6Departamento de Biología Molecular y Genómica, Instituto de Genética Humana “Dr. Enrique Corona Rivera”, Centro Universitario de Ciencias de la Salud (CUCS), Universidad de Guadalajara (UdeG), Guadalajara 44340, Jalisco, Mexico; beligomezmeda@gmail.com; 7Laboratorio de Inmunofarmacología, Centro Universitario de Ciencias Exactas e Ingenierías, Universidad de Guadalajara (UdeG), Guadalajara 44430, Jalisco, Mexico; ampueblap@yahoo.com.mx; 8Facultad de Medicina e Ingeniería en Sistemas Computacionales de Matamoros, Universidad Autónoma de Tamaulipas, Ciudad Victoria 87300, Tamaulipas, Mexico; valeriaperaltaleal@yahoo.com.mx

**Keywords:** colorectal neoplasms, *KRAS* gene, polymorphism, single nucleotide, in silico analysis, genetic variation, genetic predisposition to disease, microRNAs

## Abstract

Colorectal cancer (CRC) is a major global health challenge and one of the top 10 cancers in Mexico. Lifestyle and genetic factors influence CRC development, prognosis, and therapeutic response; identifying risk factors, such as the genes involved, is critical to understanding its behavior, mechanisms, and prognosis. The association between *KRAS* gene variants (rs8720 and rs12587) and CRC in the Mexican population was analyzed. We performed in silico analysis and analyzed 310 healthy individuals and 385 CRC patients using TaqMan assays and real-time PCR. The CC and GG genotypes of rs8720 and rs12587 were identified as CRC risk factors (*p* < 0.05). The CC and TC genotypes of the rs8720 were associated with rectal cancer, age over 50 years, moderately differentiated histology, and advanced cancer stage. TG and GG genotypes of the rs12587 variant were a risk factor in the CRC group, in patients with stage I–II, males, and stage III–IV non-chemotherapy response. The *TG* haplotype is protected against CRC. The combined CCGG genotype was linked to CRC risk. In silico analysis revealed that the rs12587 and rs8720 variants could influence KRAS gene regulation via miRNAs. In conclusion, rs8720 and rs12587 variants of the *KRAS* gene were associated with CRC risk and could influence *KRAS* regulation via miRNAs.

## 1. Introduction

CRC is defined as the uncontrolled hyperproliferation of glandular epithelial cells located in the colon and rectum due to genetic or epigenetic changes, which allows for the gradual formation of a benign adenoma, which can become cancerous and metastasize, via molecular mechanisms such as microsatellite and chromosomal instability and serrated neoplasia [1,2,3]. CRC represents the most frequent neoplasm of the digestive tract, and, according to the latest epidemiological data provided by the Global Cancer Observatory (GLOBOCAN), it is the third most frequently diagnosed neoplasm and has the second highest mortality rate [4]. Although the classic risk factors for developing this disease are lifestyle, diet, family history, and chronic inflammation, CRC is a multifactorial and highly heterogeneous disease, and genetic and environmental factors play a large role in its appearance, development, and progression. Therefore, these factors that determine the risk of developing the three known types of CRC: sporadic, hereditary, and colitis-associated [1]. The initiation, promotion, and tumor progression of this neoplasm are determined by the presence of irreversible damage to the genetic material of the epithelial cells of the colon, which promotes deregulation in various molecular signaling pathways, inducing these cells to appear abnormal. These modifications include pathways such as MAPK, Pi3K/Akt, Hedgehog, ErbB, JNK, and BMP, among others [1,2,3].

In some of these deregulated pathways in CRC, a protein of great importance in cell signaling is the KRAS protein, which is encoded by homonymous gene participates. KRAS is part of a RAS-dominated family of oncogenes; this is a series of proteins with GTPase activity. Their relationship has been demonstrated in the appearance of up to 25% of various types of human cancers, 85% of which correspond to genetic alterations present in *KRAS*. These alterations have been found in up to 98% of pancreatic ductal adenoma cases, 52% of CRC cases, and approximately 30% of lung adenocarcinomas [5,6].

The most frequent alterations found in *KRAS* are in codons 12, 13, 59, or 61, of which 97% correspond to changes or alterations to codon 12. In addition, it has been shown that *KRAS* alterations in CRC are associated with poor prognosis and resistance to treatment [7,8].

Even before the start of pharmacological therapy in metastatic cases, it is common in clinical practice to routinely perform a test for alterations in *KRAS*, since the presence of alterations in this gene also implies the provision of targeted and precise therapies [7,8].

In addition to the typical genetic alterations in *KRAS*, it has recently been shown that genetic variants, mainly single-nucleotide genetic variants (SNVs) located at microRNA (miRNA) binding sites in the 3′UTR region of *KRAS,* play an important role in the regulation of this gene; therefore, they could be associated in various ways with tumor promotion [9]. During the splicing process, the 3′UTR region is not eliminated, and this is a part of mature mRNA located downstream from the last exon. It remains unaffected by the splicing process. Therefore, the variants analyzed in this study are present in mature mRNA and may play an important role in gene expression regulation via interactions with miRNAs [10]. Several studies have shown the association between SNVs present in the 3′UTR region of the *KRAS* gene and various types of cancer, including breast cancer, Wilms tumors, colorectal cancer, and glioma, among others [11,12,13,14,15,16,17]. According to the NIH SNP database, SNV rs12587 is located at chr12:25205894 (GRCh38.p14) and represents T > G transversion, whereas rs8720 is located at chr12:25206009 (GRCh38.p13) and entails the transition T > C. On the one hand, SNV rs12587 has been associated with Wilms tumor [12] and glioma [18], but it has not been associated with CRC [13]. Meanwhile, the rs8720 variant has been associated with being a risk factor for developing CRC in the Chinese population [19].

In this study, we experimentally and in silico analyzed two SNVs, rs12587 and rs8720, both located in the 3′UTR region of the *KRAS* gene, a crucial regulatory region for gene expression. Numerous studies indicate that the post-transcriptional regulation of *KRAS* is mediated in this region by various miRNAs, suggesting that these variants may potentially interfere with its regulation [9,12,14,15]. These variants have been studied and associated with different types of cancer in diverse populations but not in the Mexican population, where studies on the association between KRAS gene variants and colorectal cancer are limited.

## 2. Material and Methods

### 2.1. Experimental Subjects

The study was carried out at the Centro de Investigación Biomédica de Occidente, Instituto Mexicano del Seguro Social and was approved by the local ethics committee (CLIES #1305) with the registration number R-2022-1305-081. All the procedures performed in the study were in accordance with the 1964 Declaration of Helsinki, and the participants provided written information. In all, 385 CRC patients with clinically and histologically confirmed CRC and 310 controls were included.

### 2.2. Variant Analysis

Using TaqMan probes designed and validated via Applied Biosystem (Thermo Fisher Scientific, Waltham, MA, USA) and real-time PCR (qPCR), the rs8720 and rs12587 variants were analyzed. The following probe sequences were used: rs8720 [VIC/FAM] 5′-ACATTACTACACAATTATCAAGAAA[T/C]CATTACTTTTTGACAAATGGAAATC-3′ (4351379 C_189578752_10) and rs12587 [VIC/FAM] 5′-GTTTCATTTTATGACAGCTATTCAG[G/T]TTCTCAATGCAGAATTCATGCTATCT-3′ (4351379 C__12104199_10). The 96-well plates containing the probe, buffer, and DNA samples with a final volume of 10 μL were read on a C1000 touch Thermal Cycler, CFX96 Real-Time PCR System (BIO-RAD, Berkeley, CA, USA). As an internal control, 10% of the reactions were analyzed twice to observe concordance in all analyzed samples.

### 2.3. In Silico Analysis

In silico analysis of the rs8720 and rs12587 SNVs and their impact on modifications in miRNA target sites was conducted. Polymorphisms in microRNAs (miRNAs) and their target sites (PolymiRTS Database version 3.0; https://compbio.uthsc.edu/miRSNP/; accessed on 20 June 2023), miRNA SNP version 3 (http://bioinfo.life.hust.edu.cn/miRNASNP; accessed on 20 June 2023), and MicroRNA Target Prediction Database (miRDB; https://mirdb.org; accessed on 20 June 2023) databases were utilized to investigate the impact of the SNVs on miRNA binding to *KRAS*, a potential target gene. Subsequently, a review was conducted using miRTarBase version 9.0 (https://mirtarbase.cuhk.edu.cn/~miRTarBase/miRTarBase_2022/php/index.php; accessed on 20 June 2023) to ascertain which miRNAs had been experimentally validated for their interaction or had bound with *KRAS*.

These databases are widely used in combination with miRNA target prediction. The targets were evaluated using bioinformatics tools that analyze massive quantities of sequencing data. Machine learning methodologies and algorithms are employed to predict the potential target genes of miRNAs. Additionally, prediction tools were utilized to analyze how SNVs can modify miRNA binding sites, potentially affecting miRNA–mRNA interaction. PolymiRTS works by integrating annotation data from databases such as UCSC and other sites focused on miRNA analysis and study, as well as incorporating large-scale experiments such as GWAS and CLASH. On the other hand, miRNASNP examines the gain or loss of miRNA target sites based on different alleles of the variants. To achieve this, the software utilizes target prediction tools such as TargetScan and miRmap. These computational tools are used to predict and evaluate potential miRNA targets while considering specific allelic variations. By leveraging these prediction algorithms, the software assesses the impact of genetic variants on miRNA target selection patterns and uncover their potential impact on gene expression regulation. In addition, miRDB analyzes and predicts miRNA target sites in the 3′UTR region of mRNA by integrating a machine learning model based on support vector machines (SVMs) and high-throughput training datasets. Furthermore, miRTarBase operates by analyzing miRNA-Target Interactions (MITs) and subsequently seeking their validation via various experimental methods [PolymiRTS Database version 3.0; https://compbio.uthsc.edu/miRSNP/; miRNA SNP version 3 http://bioinfo.life.hust.edu.cn/miRNASNP; MicroRNA Target Prediction Database, miRDB; https://mirdb.org, and miRTarBase version 9.0, https://mirtarbase.cuhk.edu.cn/~miRTarBase/miRTarBase_2022/php/index.php; accessed on 20 June 2023].

### 2.4. Co-Expression Analysis

Subsequently, a co-expression analysis of the previously filtered miRNAs was performed using the DeepMap Portal (Broad Institute; https://depmap.org/portal/; accesed on 20 June 2023), a valuable resource providing open access to analytical tools and gene expression databases. The tools for miRNA–mRNA binding prediction were utilized to investigate co-expression patterns. For the miRNAs and *KRAS* mRNA, normalized log2 values (relative to ploidy + 1) from the Copy Number Public 23Q2 consortium were employed. This analysis focused on samples from colorectal adenocarcinoma tumors.

### 2.5. KRAS Gene Expression Levels

In this study, in silico analyses were conducted to investigate the average gene expression levels of *KRAS* in colon adenocarcinoma (COAD) and rectal adenocarcinoma (READ) samples. The analysis was performed using the Gene Expression Profiling Interactive Analysis (GEPIA) tool (http://gepia.cancer-pku.cn; accessed on 20 June 2023), which integrates and analyzes data from the Cancer Genome Atlas (TCGA) and Genotype–Tissue Expression (GTEx) repositories. For this analysis, average expression data of *KRAS*, previously normalized to the logarithmic scale log2 (TPM + 1), were compared between the 275 tumor samples (COAD) and 349 normal samples as well as 92 tumor tissue samples (READ) and 318 healthy tissue samples. A significance threshold of *p* < 0.01 was set.

A comparative analysis was conducted to examine the average expression levels of *KRAS*, segregated by tumor stage, in the COAD and READ samples. Additionally, an analysis of overall survival and disease-free survival was performed for both types of neoplasms.

### 2.6. Statistical Analysis

The frequencies of the clinicopathologic variables, genotypes, and alleles were expressed using percentages. The genotypes observed and expected from the control group were compared using the chi-square test for calculating the Hardy–Weinberg equilibrium (HWE). The genotype association was analyzed using odds ratios and binary logistic regression in SPSS Statistic Base 24 (Chicago, IL, USA). The SHEsis online version of the program was used to analyze pairwise linkage disequilibrium (D’) and haplotype frequency [20]. Kaplan–Meier analysis was utilized for survival analysis in silico. A significance threshold of *p* =< 0.01 was set.

## 3. Results

### 3.1. Clinical and Demographic Characteristics of the Study Groups

The demographic variables of the study groups are described in Table 1. The average age in the group of patients with CRC was 59.86 ± 11.65 years; in the control group, the average age was 59.19 ± 13.56. The distribution by sex did not show statistically significant differences (*p* > 0.05); in the CRC group, 55% were men compared to 51% in the control group. There were no significant differences in the variables of tobacco and alcohol consumption (*p* > 0.05).

The frequency clinical variable most relevant in the CRC group was the presence of rectal type (51%), stage III (42%), moderate differentiated histology (82%), and negative lymph node metastasis (59%) (Table 2).

### 3.2. Analysis of the rs8720 and rs12587 Variants of the KRAS Gene in Patients and Control Groups

The CC genotype and recessive model (odds ratio [OR] 1.72 [95% confidence interval (CI) 1.24–2.39], *p* = 0.001), as well as the C allele frequency (OR 1.29, 95%CI 1.04–1.60, *p* = 0.019) of the rs8720 variant, were CRC risk factors. The GG genotype and recessive model of the rs12587 variant (OR 1.48, 95% CI 1.03–2.13, *p* = 0.040) were CRC risk factors (Table 3).

### 3.3. Association Analysis of the rs8720 and rs12587 Variants on the KRAS Gene with the Clinical Variables in the CRC Group

In the CRC group, the CC and TC genotypes of the rs8720 variant were associated with being risk factors in patients at stages I–II (OR 1.58, 95% CI 1.03–2.42, *p* = 0.044), more than 50 years old (OR 1.62, 95% CI 1.01–2.61, *p* = 0.044), rectal type (OR 1.7, 95% CI 1.12–2.58, *p* = 0.014), rectal at an advanced stage III–IV (OR 1.92, 95% CI 1.92–3.2, *p* = 0.017), and rectal and moderate differentiated histology (OR 1.92, 95% CI 1.22–3.01, *p* = 0.006). The GG and TG genotype of the rs12587 variant was associated with being risk factors for CRC stage I–II (OR 2.18, 95% CI 1.28–3.71, *p* = 0.004), male gender with I–II (OR 2.08, 95% CI 1.11–3.8, *p* = 0.030), and III–IV stage with a non-response to chemotherapy (OR 1.8, 95% CI 1.12–2.92, *p* = 0.01) (Table 4).

### 3.4. Haplotypes Analysis of rs8720 and rs12587 Variants of the KRAS Gene in the Studies Groups

The comparisons between the studied groups showed statistically significant differences in terms of haplotype frequency: TG (OR 0.50, 95% CI 0.37–0.68, *p* = 0.0001) (Table 5). The linkage disequilibrium of the vrs8720 and rs12587 variants showed D’ 0.35 and r’ = 0.11 (*p* = 0.0001) in the control group.

### 3.5. Genotype Combination Analysis of the rs8720 and rs12587 Variants of the KRAS Gene in CRC and Control Groups

The genotype combinations from the CRC and control group were found to be statistically different: CCGG (OR 2.4, 95% CI 1.46–3.9, *p* = 0.0005), TCGG (OR 0.30, 95% CI 0.18–0.54, *p* = 0.0002), and TTTG (OR 0.50, 95% CI 0.33–0.96, *p* = 0.049) (Table 6). They did not show statistically significant differences in terms of the demographic and clinical pathological characteristics of the group of patients with CRC.

### 3.6. Comparative Analysis of the Allelic Frequencies of rs8720 and rs12587 KRAS Variants of the Mexican Population with Different Populations

The frequencies of the C (rs8720) and G (rs12587) alleles of the *KRAS* gene variants in the control group were statistically different when compared with groups from different world populations (*p* < 0.05) except for the admixed Ashkenazi Jewish and Latino group (Figure 1). Allelic frequencies from other populations were taken from Ensembl, consulted in May 2023 (https://www.ensembl.org/Multi/Search/Results?q=rs8720;site=ensembl_all and https://www.ensembl.org/Multi/Search/Results?q=rs12587;site=ensembl_all; accessed on 20 June 2023).

### 3.7. Analysis In Silico

#### 3.7.1. miRNAs Targeting the Genomic Regions of SNPs rs8720 and rs12587

Using in silico tools, filtrations were performed to identify miRNAs whose binding sites could potentially be modified by the presence of the alleles of the rs8720 and rs12587 variants. It was observed that the C allele of rs8720 allows for binding to three different miRNAs, while the T allele shows affinity with four of them, including one previously validated experimentally. On the other hand, it was observed that the G allele of rs12587 promotes binding to six distinct miRNAs, while the T allele only binds to one of them (has-miR-4328), which was experimentally (has-miR-4328) validated among the filtered miRNAs (Figure 2).

##### 3.7.2. Analysis of miRNA/mRNA KRAS Expression Profiles

In the analysis performed using the DeepMap portal, we observed the expression profiles of each miRNA interacting with the rs8720 and rs12587 variants as well as the expression profile of *KRAS* mRNA in the colorectal adenocarcinoma tissues (Appendix A: Expression analysis of KRAS miRNA). The expression analyses are shown in Appendix A, and the correlation values are presented in Table 7.

###### 3.7.3. mRNA Expression Analysis in COAD and READ Samples

mRNA analysis in the COAD and READ samples observed a higher mean expression in the COAD tumor tissues (T = 14.34, N = 11.04) and READ (T = 14.34, N = 10.07). When the mRNA expression of *KRAS* was analyzed by stage in the COAD and READ samples, no significant differences were found between stages I, II, III, and IV (COAD; F = 0.647, Pr[>F] = 0.586) (READ; F = 1.77, Pr[>F] = 0.161). Additionally, survival analysis was conducted on tumor samples regarding *KRAS* mRNA expression, and no statistically significant differences were observed between the tumor samples and healthy tissue samples (C) (COAD; OS: HR = 0.9, *p* = 0.67; DFS: HR = 1.2, *p* = 0.53), (READ; OS: HR = 0.76, *p* = 0.58; DFS: HR = 0.67, *p* = 0.4) (Figure 3).

From the miRTARBase database (https://mirtarbase.cuhk.edu.cn/~miRTarBase/miRTarBase_2022/php/index.php; accessed on 20 June 2023), a list of genes potentially regulated by hsa-miR-4328 was accessed. Additionally, using the DAVID Bioinformatics resources database (https://david.ncifcrf.gov/home.jsp; accessed on 20 June 2023), a pathway of the genes regulated by this miRNA that are involved in cellular proliferation pathways, apoptosis, and other cancer-related pathways was reviewed (Figure 2).

## 4. Discussion

In Mexico, CRC is the third most common cause of cancer among the public and has the second most common mortality rate in both men and women [4,21]. Its highest frequency has been observed in subjects from approximately 50 years of age, reaching a maximum peak at 88 years of age [21]. This is consistent with the average age data in this study. However, different risk factors related to the presence of CRC in this respect have been noted; we observed a frequency with a similar proportion of colon and rectal type, a high frequency of moderately differentiated adenocarcinoma histology, and advanced stage III.

In the process of cellular proliferation and differentiation, the participation of different cellular signaling pathways has been observed, and one of these pathways of interest is the *KRAS* signaling pathway. It is known that the activation of KRAS stimulates the participation of signaling pathways MAP kinase and PI3K-AKT-mTOR as well as the pathways that participate in invasion and metastasis (TIAM1-RAC and RAL). Furthermore, different frequencies of mutations in the *KRAS* genes have been reported in tumors, with the incidence being high in CRC tumors (around 40%) and found mainly in the coding region of the P-loop coding protein (codon 12 and 13) [22]. The gene that codes for the KRAS protein is regulated in the 3′UTR promoter region by small non-coding RNAs approximately 20 nucleotides in length, called microRNAs (miRNA), which participate in different processes, such as proliferation, migration, invasion, and tumor development. In CRC, the participation of miRNAs in the regulation pathways can be dual, with them acting as tumor suppressors (miR-143) and oncogenes (miR-21) [23].

These are altered in cancer and have been associated with different gene variants [10]. The literature contains few association studies on the rs8720 variant; only one study, conducted in a Chinese population, has associated it with susceptibility to the risk of developing CRC [19].

Associating the variant with the TT genotype and the T allele, the study’s results are contradictory to the findings observed in this study, where we observed that the CC genotype and the C allele of the rs8720 variant were associated with a risk of developing CRC (*p* < 0.05). This is the first study to report on the association of susceptibility with the risk of developing CRC in a Mexican population.

In the rs12587 variant, the data in the present study showed an association between the GG genotype and risk in CRC (*p* < 0.05). Only one study, which was carried out in the Chinese population with CRC and included 430 patients and the same number of controls, found no association with the rs12587 variant [13].

The importance of conducting studies in each population was evidenced by the findings observed when comparing the C (rs8720) and G (rs12587) allele variants of the *KRAS* gene in the control group of the Mexican population from this study with the control groups of other populations, with differences observed in Finnish, European, East Asian, and African American populations with both variants of the *KRAS* gene. Notably, the frequency of the T allele of both polymorphisms shows similar segregation in those populations that did not show significant differences when compared with the Mexican population. The exception is the population of Puerto Rico, where the segregation of the T allele shows an inverse behavior. This evidences this gene’s genetic heterogeneity. Latin American populations are characterized by being mestizo; therefore they are populations with high genetic diversity [24]. However, more studies are needed to verify the inverted allelic frequencies observed in rs8720 and rs12587 variants among the Puerto Rican and Mexican populations; since the data was taken from a repository, a study cannot necessarily verify its frequency.

The association analysis of the clinical variables of the CRC patients with the *KRAS* gene variants showed that being a carrier of the CC and TC genotypes of the rs8720 variant was linked with rectal cancer, an advanced age over 50 years, progression, and moderately differentiated histology. Regarding the literature, there is only one study, which analyzed 1142 patients with CRC and the same number of controls from the Chinese population, where the authors observed an association between the T allele of the rs8720 variant in CRC patients and invasion beyond the serosa, suggesting that the T allele may be correlated with progression to CRC [19]. However, the results of the analysis carried out in this study show that the C allele was the most frequent in the Mexican population, which is why the risk of the C allele with the clinical pathological characteristics of the group of patients with CRC is evident.

Notably, the GG and GT genotypes of the rs12587 variant had statistical differences when compared for male gender and stages I–II and nonresponse to chemotherapy in stages III–IV. There is only one study in the literature where the association between the rs12587 variant and CRC in the Chinese population has been analyzed; however, the authors found no association [13]. Although no existing studies support these findings, these confounding factors show that this stratification is important due to its contributing to differences in the rs8720 and rs12587 variants and their associations with CRC risk.

It has been observed that the synergistic effect of RAS and the mTOR complex and the PI3K/AKT and ERK pathways has an important function in cell survival and aging. In fact, age is an important factor in the formation of cancer. It has been shown that, based on the time of diagnosis, CRC can be divided into two groups: early (before the age of 50) or late (after the age of 50) [25].

It is worth noting that, although CRC is a multifactorial entity and does not indicate that only one gene is responsible for originating metastasis, different studies have shown that the most common metastasis in CRC is in the liver and lung. Therefore, it has been shown that lung metastasis is more frequent than rectal cancer and overall survival is greater than that of patients with colon cancer. However, much remains to be known about the pathways involved in the development of metastasis of CRC tumor cells. Different RNAs of specific genes in circulating tumor cells in the bloodstream of CRC patients have been shown to be associated with cell motility, apoptosis, cell signaling and interaction, and their connection to neutrophils, all of which play important roles in the development of metastases [25].

Moreover, CRC patients with *KRAS* mutations have been shown to have an inferior response to most KRAS-targeted therapies, so the CRC consortium has suggested that a comprehensive understanding of molecular interactions in pathways is important for mutant CRC signaling in KRAS and that the integration of multi-omics data (genome, transcriptome, epigenome, metabolome, and immunome) can help to understand and propose the development of combination therapies with potential therapeutic use suitable for *KRAS* mutant CRC [7].

Although much remains unknown about the molecular mechanisms of the *KRAS* gene, it has recently been shown that variants in the 3′UTR region of the gene do not permit the binding of gene regulatory molecules, such as the microRNA-driven epigenetic regulation of *KRAS* expression, further increasing the complexity of the known *KRAS* biology. The heterogeneous group of miRNAs is composed of small, single-stranded non-coding RNA molecules. Currently, several miRNAs (let-7, miR-193, miR-143, miR-18a) have been identified that target the *KRAS* 3′UTR region, leading to *KRAS* mRNA degradation and/or repression of *KRAS*. Consistent with their function, *KRAS*-targeting miRNA levels were found to be decreased in CRC, highlighting miRNAs as potential diagnostic or prognostic biomarkers, either as single factors or in miRNA panels [7]. This can contribute to the development of cancer. Additionally, the 3′UTR of the *KRAS* gene helps regulate it by disrupting complementary sites, which promotes tumor progression. The variant alleles of rs8720 and rs12587 are probably located in the target sites of cell recognition and modulation and, consequently, may influence the imbalance of *KRAS* and the survival of cancer cells [7,11,14,25].

The *KRAS* variants analyzed in this study were not shown to be in linkage disequilibrium. In the study, in the groups analyzed, the frequent haplotype rs8720 and rs12587 observed to act as a protective factor against susceptibility to the development of CRC was TT (present in 20% of the controls and 11% of the patients with CRC). Unfortunately, no study on BC has analyzed this association. Thus, the combination of the two *KRAS* variants is important information that identifies the haplotypes that confer protection against developmental susceptibility to CRC.

On the other hand, the analysis of the combination of genotypes of the analyzed variants of the *KRAS* gene (rs8720 and rs12587) showed the CCGG combination as a risk factor for the susceptibility of developing CRC, indicating that, in patients with CRC, both risk alleles must be present. It should be noted that more population studies are necessary to demonstrate this association.

### In Silico Analysis

Through the use of repositories, the differential expression of 13 miRNAs that interact with the rs8720 and rs12587 variants was demonstrated. It is worth mentioning that, for the 13 miRNAs associated with the rs8720 and rs12587 variants of the *KRAS* gene, their relationship with different pathologies has been analyzed via experimental studies in culture or animal models as well as in groups of patients. In this regard, hsa-miR-885-5p participates as a suppressor of the expression of this lipid receptor and sterol transporter, being associated with the regulation of fatty liver and lipoprotein metabolism [26]. A has-miR-497-3p study performed in an experimental model in rats with induced physiological left ventricular hypertrophy showed that the expression levels of miR-26b-5p, miR-204-5p, and miR-497-3p participated in autophagy regulation [26,27]. Another study has suggested that miR-497 and miR-1246 are possibly involved in the progression of hepatocarcinoma by regulating target genes [28]. has-miR-4328 can be considered a possible biomarker in acute promyelocytic leukemia [29]. In addition, it has also been found to participate in diabetic retinopathy [30]. has-miR-382-3p has been related to the regulation of spermatogenesis [31]. Meanwhile, has-miR-182-3p participates in the pathogenesis of pulmonary arterial hypertension vascular remodeling [32] as well as in the regulation of osteosarcoma through the EBF2 regulation pathway [33]. Furthermore, it was shown that the overexpression of has-miR-152-5p inhibits the progression of fibrosis in keloids [34]. Another study found this microRNA to be a potential biomarker for ST-segment elevation myocardial infarction [35]. It has been suggested that has-miR-11399 regulates interleukin 6 (IL-6), associating it with vascular events, stress, and depression [36]. On the other hand, there is a lack of studies on other miRNAs (e.g., has-miR-6512-5p, has-miR-597-3p, has-miR-5680, has-miR-551b-5p, has-miR-506-5p, has-miR-2117).

The miRNAs identified in the present study with the use of bioinformatics tools propose new lines of research in the *KRAS* gene in CRC and in other types of pathologies. It should be noted that it was found that hsa-miR-4328 miRNA regulate genes involved in various cell signaling pathways, participate in the regulation of EFGR, RET/PTC, KRAS, C-JUN, and AP1/SP1 in the MAPK pathway and the *ET1* and *PTEN* genes that participate in the PI3K-AKT signaling pathway. However, the in silico study did not show differences between mRNA expression levels and survival modification between colon and rectal cancer. Although a high expression was observed in them, it has not been ruled out that these miRNAs may be involved in the regulation of the gene, since the in silico analysis showed that at least 13 miRNAs showed an association with the alleles of the rs8729 and rs12587 variants analyzed in the present study. New studies are recommended to validate this information.

## 5. Conclusions

Our results showed that the CC genotype, C allele, and dominant (TCCC) of the rs8720 variant were associated with a risk for CRC when the controls and patients were compared. Furthermore, differences were observed in the patients with CRC stratified by CC and TC genotype of the rs8720 variant and the presence of rectal cancer, stage I–II, or rectal type to age greater than 50 years old, with moderately differentiated tumor and with stages III–IV. The rs12587 variant was also a risk factor for CRC patient group carriers of the GG genotypes when the controls and patients were compared. Differences were observed in the patients with CRC stratified by GG and TG, and stage I–II, male and stage I–II, and non-chemotherapy response to stage III–IV. The presence of TG haplotypes is an associated protective susceptibility factor in CRC. The identification of 13 miRNAs interacting with the variants analyzed in silico as well as different signaling pathways is important. More studies are needed to confirm the observed findings.

## Figures and Tables

**Figure 1 cells-12-01941-f001:**
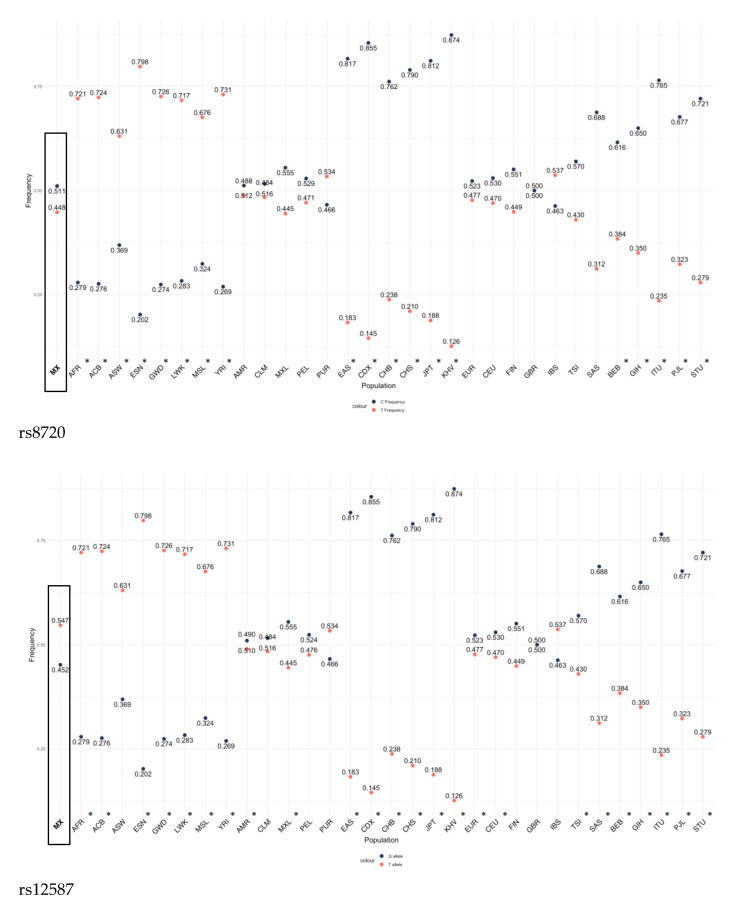
Comparison of the allelic frequency of different populations of the rs8720 and rs12587 variants of the *KRAS* gene with the frequency of the Mexican population described in this study. * *p* < 0.05. MX (Mexican population, analyzed in the present study), ACB (African Caribbean in Barbados), AFR (African), ASW (African Caribbean in Barbados), ESN (Esan in Nigeria), GWD (Gambian in Western Division), LWK (Luhya in Webuye, Kenya), MSL (Mende in Sierra Leone), YRI (Yoruba in Ibadan, Nigeria), AMR (American), CLM (Colombian in Medellin), MXL (Mexican Ancestry in Los Angeles, California), PEL (Peruvian in Lima), PUR (Puerto Rican in Puerto Rico), EAS (East Asian), CDX (Chinese Dai in Xishuangbanna), CHB (Han Chinese in Bejing), CHS, (Southern Han Chinesse), JPT (Japanese in Tokio), KHV (Kinh in Ho Chi Minh City, Vietnam), EUR (European), CEU (Utah residents with Northern and Western European ancestry), (FIN) Finnish in Finland, GBR (Birtish in England and Scotland), IBS (Iberian Populations in Spain), TSI (Toscani in Italy), SAS (South Asian), BEB (Bengali in Blagladesh), GIH (Gujarati Indian in Houston), ITU (Indian Telugu in the UK), PJL (Punjarbi in Lahora, Pakistan), STU (Sri Lankan Tamil in the UK). The frequencies of the other populations were taken from https://www.ensembl.org/Multi/Search/Results?q=rs8720;site=ensembl_all and https://www.ensembl.org/Multi/Search/Results?q=rs12587;site=ensembl_all; accessed on 20 June 2023.

**Figure 2 cells-12-01941-f002:**
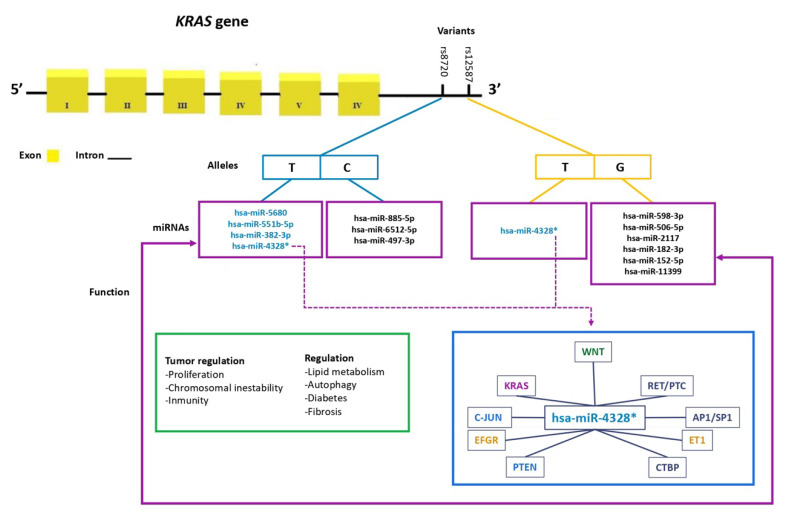
miRNA binding sites at the rs8720 and rs12587 variants of the *KRAS* gene and their function in tumor regulation. (*) miRNA has been experimentally validated; it was observed that hsa-miR-4328 regulates genes involved in various cellular signaling pathways, such as *EFGR, RET/PTC, KRAS, C-JUN, AP1/SP1* in the MAPK pathway and the genes *ET1* and *PTEN* in the PI3K-AKT signaling pathway.

**Figure 3 cells-12-01941-f003:**
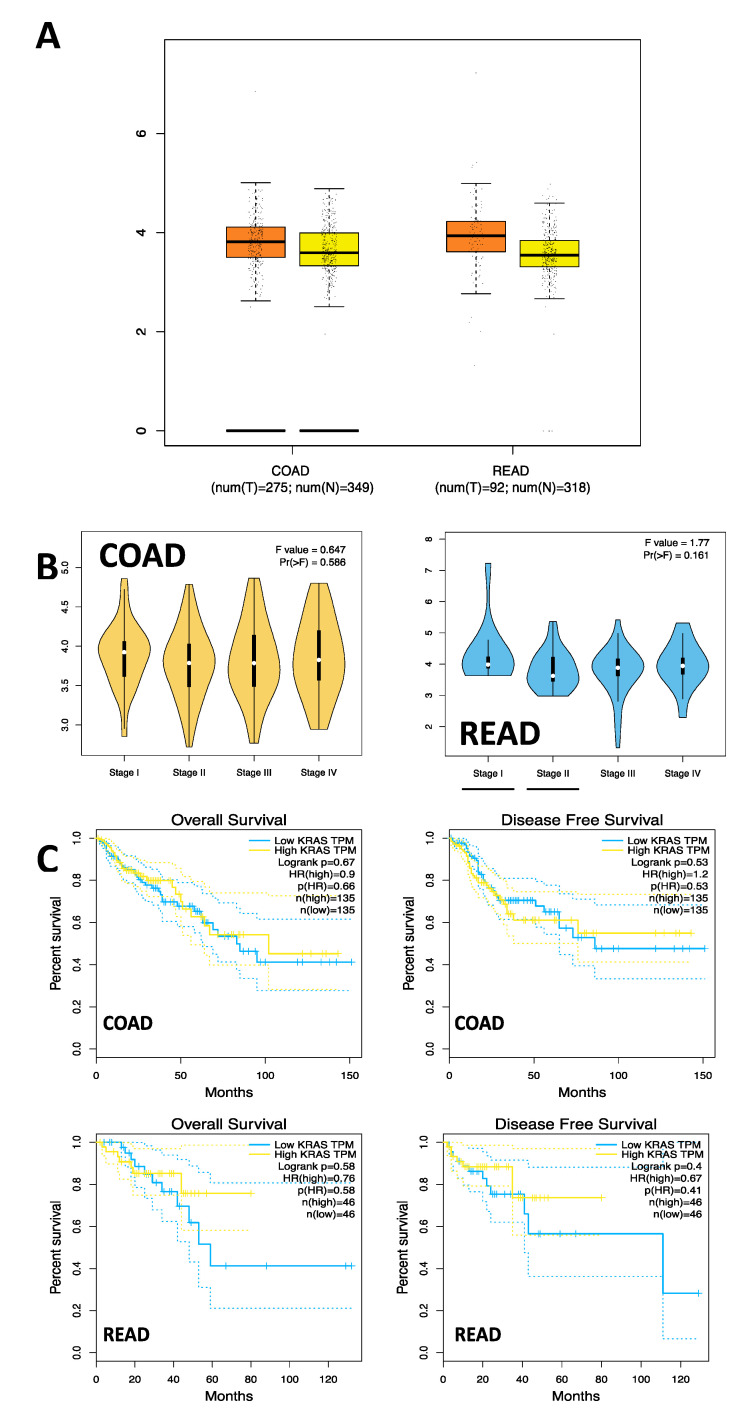
mRNA Expression and survival analysis in COAD and READ Samples. (**A**) mKRAS/hsa-miR4328* (r = 0.018, *p* = 0.871). (**B**) mKRAS/hsa-miR-506-50 (r = 0.071, *p* = 0.526). (**C**) mKRAS/hsa-miR-597-3p (r = −0.037, *p* = 0.743).

**Table 1 cells-12-01941-t001:** Demographic data for the study groups.

	CRC Patients ^(n = 385)^		Controls ^(n = 310)^		*p* Value
**Age** (years, average ± SD)	59.86±	11.65	59.19±	13.56	0.484 *
	n	%	n	%	
≤49 years	(72)	19.0	(66)	21.0	0.502 **
≥50 years	(313)	81.0	(244)	79.0	
**Sex**					
Male	(211)	55.0	(157)	51.0	0.309 **
Female	(174)	45.0	(153)	49.0	
**Tobacco consumption**					
Yes	(126)	33.0	(106)	34.0	0.744 **
No	(259)	67.0	(204)	66.0	
**Alcohol consumption**					
Yes	(133)	35.0	(85)	27.0	0.053 **
No	(252)	65.0	(225)	73.0	

SD (standard deviation); * Student’s *t*-test; ** Chi-square test.

**Table 2 cells-12-01941-t002:** General description of clinical pathological characteristics of CRC group.

CRC Patients ^(n = 385)^					
	(n)	%		(n)	%
**Location**			**Histological classification Adenocarcinoma**		
Rectum	(196)	51	Moderate differentiated	(317)	82
Colon	(189)	49	No differentiated	(18)	5
**Stage**			Differentiated	(37)	10
I	(8)	2	Mucinous	(13)	3
II	(127)	33	**Lymph node metastasis**		
III	(162)	42	Positive	(158)	41
IV	(88)	23	Negative	(227)	59

**Table 3 cells-12-01941-t003:** Genotype and allelic distribution of the rs8720 and rs12587 variants of the *KRAS* gene in CRC patients and a control group.

Variants			CRC		Controls *		OR	95%(CI)	*p* Value
**rs8720**	Model	Genotype	(n = 385)	%	(n = 310)	%			
		TT	(88)	23	(74)	24	1.0		
	Codominant	TC	(151)	39	(155)	50	0.64	(0.47–0.87)	0.005
		CC	(146)	38	(81)	26	1.72	(1.24–2.39)	0.001
	Dominant	TT	(83)	22	(137)	28			
		TC + CC	(294)	78	(355)	72	1.05	(0.74–1.50)	0.753
	Recessive	CC	(145)	38	(112)	23	1.72	(1.24–2.39)	0.001
		TT + TC	(232)	62	(380)	77			
		**Alleles**	**(2n = 770)**		**(2n = 620)**				
		T	(327)	0.4246	(303)	0.4887	0.77	(0.62–0.95)	0.019
		C	(443)	0.5754	(317)	0.5113	1.29	(1.04–1.60)	0.019
**rs12587**		**Genotypes**	(n = 383)	%	(n = 282)	%			
		TT	(110)	29	(86)	30	1.0		
	Codominant	TG	(165)	43	(137)	49	0.80	(0.58–1.09)	0.183
		GG	(108)	28	(59)	21	1.48	(1.03–2.13)	0.040
	Dominant	TT	(110)	29	(86)	30			
		TG + GG	(273)	71	(196)	70	1.08	(0.78–1.52)	0.619
	Recessive	GG	(108)	25	(59)	25	1.48	(1.03–2.13)	0.040
		TT + TG	(275)	75	(223)	75			
		**Alleles**	**(2n = 766)**		**(2n = 564)**				
		T	(385)	0.5026	(309)	0.5478	0.83	(0.67–1.03)	0.114
		G	(381)	0.4974	(255)	0.4522	1.19	(0.96–1.49)	0.114

OR (odds ratio), CI (confidence intervals, p value adjusted (significant < 0.05). * Hardy–Weinberg equilibrium (HWE) for the control group: variants rs8720 (chi-square test = 0.045; *p* =0.830), and rs12587 (chi-square test = 0.0804, *p* = 0.77).

**Table 4 cells-12-01941-t004:** *KRAS* gene variants (rs8720 and rs12587) and their association with the clinic–pathologic variant of the CRC group.

Variant	Genotype	Variable	OR	95%(CI)	*p* Value
**rs8720**	CC	I–II stage	1.58	(1.03–2.42)	0.044
	CC	≥50 years old and I–II stage	1.62	(1.01–2.61)	0.044
	TC	Rectal	1.7	(1.12–2.58)	0.014
	TC	Rectal and ≥50 years old	1.58	(1.01–2.5)	0.046
	TC	Rectal and III–IV stage	1.92	(1.14–3.2)	0.017
	TC	Rectal and Mod differentiate	1.92	(1.22–3.01)	0.006
**rs12587**	GG	I–II stage	2.18	(1.28–3.71)	0.004
	GG	Male and stage I–II	2.08	(1.11–3.8)	0.030
	TG	Non-chemotherapy response and III–IV stage	1.8	(1.12–2.92)	0.01

Bivariate analysis, OR (odds ratio), CI (confidence intervals, p value adjusted (significant < 0.05).

**Table 5 cells-12-01941-t005:** Haplotype frequency of the rs8720 and rs12587 variants of the *KRAS* gene in CRC and control groups.

Genotype		CRC ^(2n = 734)^		Controls ^(2n = 556)^			
rs8720	rs12587	n	%	n	%	OR 95%(CI)	*p* Value
T	T	(225)	31	(161)	29	1.1 (0.85–1.4)	0.50
T	G	(83)	11	(112)	20	0.50 (0.37–0.68)	0.0001
C	G	(286)	39	(192)	35	1.2 (0.96–1.52)	0.11
C	T	(140)	19	(91)	16	1.2 (0.90–1.61)	0.23

D’ (0.35) and r2 (0.11).

**Table 6 cells-12-01941-t006:** Genotypes combination of the studied group.

		CRC ^(2n = 367)^		Controls ^(2n = 278)^			
rs8720	rs12587	n	%	n	%	OR 95%(CI)	*p* Value
CC	GG	(68)	19	(24)	9	2.4 (1.46–3.9)	0.0005
CC	TG	(40)	11	(25)	9	1.2 (0.73–2.0)	0.50
CC	TT	(36)	10	(22)	8	1.2 (0.72–2.2)	0.48
TC	GG	(21)	6	(45)	16	0.3 (0.18–0.54)	0.0002
TC	TG	(89)	24	(74)	26	0.8 (0.61–1.26)	0.55
TC	TT	(28)	8	(22)	8	0.9 (0.53–1.7)	1.0
TT	GG	(17)	4	(16)	6	0.7 (0.39–1.60)	0.64
TT	TG	(28)	7	(35)	13	0.5 (0.33–0.96)	0.049
TT	TT	(40)	11	(15)	5	1.6 (0.57–4.91)	0.47

**Table 7 cells-12-01941-t007:** Expression analyses and the correlation values of miRNA/mRNA *KRAS* expression profiles.

miRNA		(n)	Pearson	Spearman	Slope	Intercept	*p* Value (Line Regress)
**hsa-miR-4328**	(A)	82	0.018	−0.036	8.80 × 10^−3^	7.67 × 10^−1^	8.71 × 10^−1^
**hsa-miR-506-50**	(B)	82	0.071	0.159	3.47 × 10^−2^	7.43 × 10^−1^	5.26 × 10^−1^
**hsa-miR-597-3p**	(C)	82	−0.037	−0.155	−1.77 × 10^−2^	8.38 × 10^−1^	7.43 × 10^−1^
**hsa-miR-152-5p**	(D)	82	0.046	0.015	1.74 × 10^−2^	1.01 × 10^0^	6.80 × 10^−1^
**hsa-miR-182-3p**	(E)	82	0.057	0.150	2.15 × 10^−2^	1.12 × 10^0^	6.13 × 10^−1^
**hsa-miR-2117**	(F)	82	0.040	0.068	1.15 × 10^−2^	1.01 × 10^0^	7.24 × 10^−1^
**hsa-miR-6512-5p**	(G)	82	−0.116	0.033	−2.22 × 10^−2^	1.05 × 10^0^	2.98 × 10^−1^
**hsa-miR-885-5p**	(H)	82	−0.124	−0.078	−3.86 × 10^−2^	9.96 × 10^−1^	2.69 × 10^−1^
**hsa-miR-497-3p**	(I)	82	0.162	−0.015	5.88 × 10^−2^	8.24 × 10^−1^	1.45 × 10^−1^
**hsa-miR-6512-5p**	(J)	82	−0.116	0.033	−2.22 × 10^−2^	1.05 × 10^0^	2.98 × 10^−1^
**hsa-miR-551b-5p**	(K)	82	0.023	0.084	8.24 × 10^−3^	1.10 × 10^0^	8.35 × 10^−1^
**hsa-miR-382-3p**	(L)	82	0.109	0.085	3.81 × 10^−2^	9.62 × 10^−1^	3.29 × 10^−1^
**hsa-miR-5680**	(M)	82	−0.173	−0.152	−6.98 × 10^−2^	1.13 × 10^0^	1.21 × 10^−1^

## Data Availability

The data that support the findings of this study are available from the corresponding author upon reasonable request. Data available on request due to privacy/ethical restrictions.

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
