# Peer review of "Association of the rs8720 and rs12587 KRAS Gene Variants with Colorectal Cancer in a Mexican Population and Their Analysis In Silico"

_cells, 2023, doi:10.3390/cells12151941_

Round 1

Reviewer 1 Report

Table 1 data is not clear in columns. I cant understand which one is SD and Which one is mean.

In figure 3 names of miRNAs is not visible on graph. As Table 7 shows the same data, i suggest figure 3 to move into supplementary file with higher quality of presentation.

The role of risk factors is not analysed in regression analysis and I did not understand why they are presented in demographic table.

First sentence of discussion need reference.

I recommend drowing a normogram graph for better predictions of risk factors with R.

Author Response

We appreciate your attention to our manuscript entitled “Association of the rs8720 and rs12587 KRAS gene variants with colorectal cancer in a Mexican population and their analysis in silico” cells-2506157. By Martha Patricia Gallegos-Arreola, Asbiel Felipe Garibaldi-Ríos, José Israel Cruz-Sánchez, Luis Eduardo Figuera, Carlos Alberto Ronquillo-Carreón, Mónica Alejandra Rosales-Reynoso, Belinda Claudia Gómez-Meda, Irving Alejandro Carrillo-Dávila, Ana María Puebla-Pérez, Héctor Montoya-Fuentes, Valeria Peralta-Leal, and Guillermo M. Zúñiga-González*. We submit again to your editorial authority for publication in Cells.

All reviewers´ comments have been considered and pertinent changes have been incorporated into this version of the manuscript. The changes made to the document are highlighted in yellow in the text and are detailed below.

Reviewer 1

Table 1 data is not clear in columns. I can’t understand which one is SD and Which one is mean.

The comments were considered and modifications are included in Table I, this was modified in the text line 197

In Figure 3 names of miRNAs is not visible on graph. As Table 7 shows the same data, I suggest figure 3 to move into supplementary file with higher quality of presentation.

We took into consideration the feedback and transferred Figure 3 to the supplementary materials section as Figure S1 for enhanced visualization.

The role of risk factors is not analyzed in regression analysis and I did not understand why they are presented in demographic table.

First sentence of discussion need reference.

I recommend drawing a normogram graph for better predictions of risk factors with R.

The reference to the first sentence of the discussion has been included. Linear regression analysis was not performed to demonstrate the association of the clinical variables of the patients with CRC with the rs8720 and rs12587 variants. However, we considered the binary regression analysis to demonstrate this association (Table 4).

Reviewer 2 Report

The title of the manuscript is remarkable. English language has good quality. Figures need some changes. Tables have acceptable quality. Main text need some modifications. There are some modifications that need to be exerted in the citations.

1. Please rewrite the section "Abstract" based on order below:

Talk briefly about:

+ colorectal cancer

+ colorectal cancer in mexico

+ various risk factors of colorectal cancer

+ the importance of assessment of genes in diagnosis, treatment and prognosis of colorectal cancer

+ the importance of your work

+ material and methods

+ results

+ conclusion

2. Why Line 43-44 does not have proper

reference(s)?

3. All multipple and middle sentence references in all over the manuscript should be reformed

4. Why line 70-81 in page 2 has no reference?

5. About the section results:

Please devide this section into some subheadings based on your findings

6. Please make the texts in your figures more bigger so that they become obviously visible

9. About the section "Discussion"

+ Please categorize your results based on their importance from the most important one to the least. After that, discuss about each one of them one by one.

10. Please check and adjust the "Reference list" based on the regulations of reference list of journal. (Titles, doi, the name of journal and ... )

Author Response

We appreciate your attention to our manuscript entitled “Association of the rs8720 and rs12587 KRAS gene variants with colorectal cancer in a Mexican population and their analysis in silico” cells-2506157. By Martha Patricia Gallegos-Arreola, Asbiel Felipe Garibaldi-Ríos, José Israel Cruz-Sánchez, Luis Eduardo Figuera, Carlos Alberto Ronquillo-Carreón, Mónica Alejandra Rosales-Reynoso, Belinda Claudia Gómez-Meda, Irving Alejandro Carrillo-Dávila, Ana María Puebla-Pérez, Héctor Montoya-Fuentes, Valeria Peralta-Leal, and Guillermo M. Zúñiga-González*. We submit again to your editorial authority for publication in Cells.

All reviewers´ comments have been considered and pertinent changes have been incorporated into this version of the manuscript. The changes made to the document are highlighted in yellow in the text and are detailed below.

The title of the manuscript is remarkable. English language has good quality. Figures need some changes. Tables have acceptable quality. Main text need some modifications. There are some modifications that need to be exerted in the citations.

  1. Please rewrite the section "Abstract" based on order below:

Talk briefly about: + colorectal cancer, + colorectal cancer in Mexico, + various risk factors of colorectal cancer, + the importance of assessment of genes in diagnosis, treatment and prognosis of colorectal cancer, + the importance of your work, + material and methods, + results, + conclusion

The abstract was reviewed and improved, taking into account the proposed order, and focusing on the most relevant information

2.Why Line 43-44 does not have proper

reference(s)?

  1. All multipple and middle sentence references in all over the manuscript should be reformed
  2. Why line 70-81 in page 2 has no reference?
  3. About the section results:

Please devide this section into some subheadings based on your findings

  1. Please make the texts in your figures more bigger so that they become obviously visible
  2. About the section "Discussion"

+ Please categorize your results based on their importance from the most important one to the least. After that, discuss about each one of them one by one.

  1. Please check and adjust the "Reference list" based on the regulations of reference list of journal. (Titles, doi, the name of journal and ... )

The references have been included in the indicated paragraphs and appropriately corrected according to the instructions of the journal, the results have been distinguished by subtitles, and the discussion was mentioned according to the order of results. Figure 3 was changed to supplementary material (Figure S1) to ensure correct visibility of the miRNA names. Additionally, Figures 1 and 4 were expanded to improve the visibility of the text.

Reviewer 3 Report

The study is properly designed, and article is well-written, therefore I recommend to accept it subject to minor rectifications.

Please find my comments below.

Line 43. Better definition of the Colorectal cancer (CRC), allowing to distinguish it from benign tumors of the colon, should be used.

Line 71. “At microRNA binding sites”. Here, the “miRNA” abbreviation should be introduced (“At microRNA (miRNA) binding sites).

As for the study cohort, given the potential impact of the ethnic and racial component on the pathogenesis of several types of cancer (including CRC), have the authors ensured that both control and patient's groups had similar ethnic and racial composition? Especially, given the differences in the frequencies of both polymorphisms in case of the population of Puerto Rico as compared with the Mexican population, discussed further in the text?

Line 152. Table I.

“Age at diagnosis” is not correct wording to use in case of the control group

Figure 3. The text labels in X and Y axis, and the numbers on the scale, are unreadable. The same for Figure 4B.

Author Response

We appreciate your attention to our manuscript entitled “Association of the rs8720 and rs12587 KRAS gene variants with colorectal cancer in a Mexican population and their analysis in silico” cells-2506157. By Martha Patricia Gallegos-Arreola, Asbiel Felipe Garibaldi-Ríos, José Israel Cruz-Sánchez, Luis Eduardo Figuera, Carlos Alberto Ronquillo-Carreón, Mónica Alejandra Rosales-Reynoso, Belinda Claudia Gómez-Meda, Irving Alejandro Carrillo-Dávila, Ana María Puebla-Pérez, Héctor Montoya-Fuentes, Valeria Peralta-Leal, and Guillermo M. Zúñiga-González*. We submit again to your editorial authority for publication in Cells.

All reviewers´ comments have been considered and pertinent changes have been incorporated into this version of the manuscript. The changes made to the document are highlighted in yellow in the text and are detailed below.

The study is properly designed, and article is well-written, therefore I recommend to accept it subject to minor rectifications.

Please find my comments below.

Line 43. Better definition of the Colorectal cancer (CRC), allowing to distinguish it from benign tumors of the colon, should be used.

Line 71. “At microRNA binding sites”. Here, the “miRNA” abbreviation should be introduced (“At microRNA (miRNA) binding sites).

The abbreviation was added from this point onwards.

As for the study cohort, given the potential impact of the ethnic and racial component on the pathogenesis of several types of cancer (including CRC), have the authors ensured that both control and patient's groups had similar ethnic and racial composition? Especially, given the differences in the frequencies of both polymorphisms in case of the population of Puerto Rico as compared with the Mexican population, discussed further in the text?

In this regard, the following paragraph was included in the discussion, “Latino populations are characterized by being mestizo, therefore they are heterogeneous populations. However, more studies are required to verify the inverted allelic frequencies observed in rs8720 and rs12587 variants, among the Puerto Rican and Mexican populations, since the data was taken from a repository, a study cannot necessarily verify their frequency, (Lines 362-366).

Line 152. Table I. “Age at diagnosis” is not correct wording to use in case of the control group

It was corrected and kept as "age".

Figure 3. The text labels in X and Y axis, and the numbers on the scale, are unreadable. The same for Figure 4B.

Figure 3 was changed to supplementary material (Figure S1) to ensure correct visibility of the miRNA names. Additionally, Figures 1 and 4 were enhanced to improve the visibility of the text

Reviewer 4 Report

MAJOR POINTS

  • Why are these two SNPs studied.
  • What makes the studied population special?
  • It need to be explained why the miRNA targeting was conducted. What was the rationale? Where they previous reports in which those two particular SNPs (i.e., rs8720 and rs12587) were associated to miRNA binding sites in other contexts?
  • From a mechanistically point of view, explain the relevance of miRNA binding since they are not located in exons. Need to prove that the SNPs are present in the mRNA, enabling post-transcriptional miRNA-mediated effects.
  • Table 7 shows that there is no correlation (i.e. low correlation coefficients and p-values >0.05) between the expression of the selected miRNAs and mRNAs.
  • What is the connection between the Analysis of KRAS mRNA expression in COAD and READ samples with the SNP presence and/or miRNA expression.

MINOR POINTS

  • Need to provide primer and probe sequence and catalog number of the TaqMan assay
  • Which software and version was used for miRNA target prediction? Need to provide details on the "machine learning methods" and ""prediction tools" applied.
  • In Table 1, p-value for sex is incongruent with that reported in the text.
  • Move reference of Allelic Frequencies in Ensembl to Materials and Methods.

Check the tables, some words in Spanish detected

Author Response

We appreciate your attention to our manuscript entitled “Association of the rs8720 and rs12587 KRAS gene variants with colorectal cancer in a Mexican population and their analysis in silico” cells-2506157. By Martha Patricia Gallegos-Arreola, Asbiel Felipe Garibaldi-Ríos, José Israel Cruz-Sánchez, Luis Eduardo Figuera, Carlos Alberto Ronquillo-Carreón, Mónica Alejandra Rosales-Reynoso, Belinda Claudia Gómez-Meda, Irving Alejandro Carrillo-Dávila, Ana María Puebla-Pérez, Héctor Montoya-Fuentes, Valeria Peralta-Leal, and Guillermo M. Zúñiga-González*. We submit again to your editorial authority for publication in Cells.

All reviewers´ comments have been considered and pertinent changes have been incorporated into this version of the manuscript. The changes made to the document are highlighted in yellow in the text and are detailed below.

MAJOR POINTS

  • Why are these tw the points appropriately noted, were considered in the text the points appropriately noted, were considered in the texto SNPs studied.

What makes the studied population special?

  • It need to be explained why the miRNA targeting was conducted. What was the rationale? Where they previous reports in which those two particular SNPs (i.e., rs8720 and rs12587) were associated to miRNA binding sites in other contexts?
  • From a mechanistically point of view, explain the relevance of miRNA binding since they are not located in exons. Need to prove that the SNPs are present in the mRNA, enabling post-transcriptional miRNA-mediated effects.

 These points were considered and clarified at the beginning of lines 85-88 and 98-105.

  • Table 7 shows that there is no correlation (i.e. low correlation coefficients and p-values >0.05) between the expression of the selected miRNAs and mRNAs.
  • What is the connection between the Analysis of KRAS mRNA expression in COAD and READ samples with the SNP presence and/or miRNA expression.

 In this regard, as mentioned in the text, no differences were observed between the expression levels of the miRNAs target prediction in colon and rectal tissues, however, a high expression was observed in them (lines 301-303). A new paragraph was considered in the discussion. However, it is not ruled out that these miRNAs may be involved in the regulation of the gene, as the in silico analysis showed that at least 13 miRNAs showed association with the alleles of the rs8729 and rs12587 variants analyzed in the present study. Further studies are need to verify these observations. (Lines 465-468).

 MINOR POINTS

  • Need to provide primer and probe sequence and catalog number of the TaqMan assay

The primer and probes sequence, catalog number, and respective ID of the probes used were added.

  • Which software and version was used for miRNA target prediction? Need to provide details on the "machine learning methods" and ""prediction tools" applied.

The versions of the software used, when applicable (highlighted in yellow), have been included. Furthermore, a paragraph describing the functionality of each software employed for miRNA-target prediction has been integrated

  • In Table 1, p-value for sex is incongruent with that reported in the text.
  • Move reference of Allelic Frequencies in Ensembl to Materials and Methods.

Data from Table 1 was corrected appropriately and the reference to allelic frequencies was moved to the material and methods section.

Check the tables, some words in Spanish detected

The detected Spanish words in the tables were corrected and highlighted in yellow.

ADDITIONAL CORRECTIONS REQUESTED BY THE EDITOR: 

Supplementary Materials, Author Contributions, Funding, Institutional Review Board Statement

Informed Consent Statement, Data Availability Statement, Acknowledgments, Conflicts of Interest

The points appropriately noted were considered in the text.

We appreciate the time and comments of each one of the reviewer to improve the quality of this article. We hope that this new corrected version meets the standards of the journal

Round 2

Reviewer 1 Report

Thank you for corrections.

Reviewer 2 Report

I do not have more suggestion. Thanks

Reviewer 4 Report

All the points I mentioned in my previous report were properly addressed.